# Time Constant as the Main Component to Optimize the Resistance Temperature Sensors’ Calibration Process

**DOI:** 10.3390/s24020487

**Published:** 2024-01-12

**Authors:** Maciej Klebba, Arkadiusz Frącz, Michal Brodzicki, Adrianna Rzepkowska, Mariusz Wąż

**Affiliations:** 1Faculty of Mechanical and Electrical Engineering, Polish Naval Academy, 81-127 Gdynia, Poland; a.fracz@amw.gdynia.pl (A.F.); m.brodzicki@amw.gdynia.pl (M.B.); 2Doctoral School of GMU, Gdynia Maritime University, 81-87 Morska Street, 81-225 Gdynia, Poland; a.rzepkowska@amw.gdynia.pl; 3Institute of Navigation and Hydrography, Polish Naval Academy, 81-127 Gdynia, Poland; m.waz@amw.gdynia.pl

**Keywords:** optimization, temperature, resistance sensors, calibration

## Abstract

Temperature sensor calibration in the majority of measurement laboratories is performed on the ground of measurement procedures. The procedures specify the timespan for the temperature of the tested sensor to stabilize. In practice, the fact of time constant increase above expectations can be observed. This results from the method of operation, time of use, and level of contamination of individual sensors. The paper proposes a method to optimize the calibration process of resistance temperature sensors based on the measurement of the time constant and, on this basis, to determine the sensor stabilization time during calibration.

## 1. Introduction

Temperature measurement is one of the basic measurements performed both in technological and production processes as well as during the operation of machines and devices. During the operation of marine power plants, temperature is one of the main diagnostic signals. Its proper measurement determines many decisions regarding further use of the equipment, sending it back for repair or replacing it with a new one. Accurate measurement of a physical quantity that varies over time, with a known and permissible error, requires the use of a measurement track with known dynamic properties, which equally depends on the dynamic properties of all its elements. To determine whether the influence of the dynamic properties of the measurement track is small enough, it should be compared with the dynamic properties of the measured physical quantity of the tested object. For energy facilities, in which thermal and mechanical processes dominate, changes in physical parameters such as pressures, flows, and especially temperatures are relatively slow. Thus, the impact of the dynamic properties of the measurement path on the measurement accuracy can be neglected. However, there are some exceptions, which include temperature measurements with resistance and thermoelectric sensors. The dynamic properties of measurement tracks based on these types of sensors are particularly important in automatic control when we need to identify the control object in which the dynamic properties of the measurement track may be important.

Issues related to temperature measurement are widely described in literature and scientific articles. The articles describe issues related to the optimization of the furnace temperature curve [1], a thermal conductivity sensor for gas detection [2], or a prototype of a variable temperature insert intended for the calibration of cryogenic thermometers [3]. Various techniques for calibrating temperature sensors are also described [4,5,6,7]. The articles [8,9,10] present research showing the influence of fluid flow speed on the time constant and measurement errors of temperature sensors. 

By the regulations in the Polish Army, all measuring instruments are subjected to periodic metrological inspection. Resistance temperature sensors, such as Pt100, Pt500, Cu50, etc., must be calibrated every two years. Certified laboratories perform metrological services by the principles described in the PN-EN ISO 17025 standard [11]. The calibration procedure considering resistance temperature sensors is based on the provisions of the PN-EN IEC 60751 standard [12].

The calibration procedure described in the standard [12] assumes that each resistance temperature sensor should be checked in a specific temperature range. The next test is to check the sensor’s time constant. The calibration of clean sensors is thoroughly described, tested, and verified. However, shipboard practice often deals with sensors whose covers are partially damaged, dirty, or covered with something. This affects both the response time and the measured temperature value itself.

In the article, the authors try to verify the impact of typical pollutants on basic static and dynamic parameters checked during calibration. 

## 2. Materials and Methods

### 2.1. Mathematical Model of an Ideal Sensor

The dynamic properties of temperature sensors can be determined experimentally or analytically. To analytically determine the dynamic properties, a mathematical model should be created based on knowledge of the laws of heat transfer and the structure of the sensor. Typically, several simplifications are made. To create a mathematical model, heat balance equations written in the form of partial differential equations are used (distributed constants model), or, assuming a uniform temperature distribution (infinite thermal conductivity value), the differential equations are reduced to an ordinary equation (lumped constants model). The development of computer computational techniques allows the numerical analysis of sensor properties using finite element methods. However, the values of many thermal quantities taken into account in the mathematical model may still differ from the real ones. This is why the method of experimentally identifying the elements of the measurement track and their dynamic properties, using standard signals introduced to their input, such as the Dirac pulse or jump, is widely used. 

Considering the “ideal temperature sensor”, the following simplifying assumptions are made:infinite thermal conductivity *λ* of the sensor material, which is equivalent to the same temperature throughout the entire volume of the sensor,constant heat transfer conditions by convection (α = const.),complete immersion of the sensor in the tested medium so that there is no heat exchange with another medium.No impact of the sensor on the tested medium, which means that the capacity of the medium is much larger than the thermal capacity of the sensor, and the presence of the sensor does not affect the deformation of the temperature field of the tested medium.

Considering heat flow from the point of view of thermokinetics, bodies meeting the above-mentioned conditions are called fine charges. The energy balance for unsteady thermal conditions of the sensor takes the form of the amount of heat *Q* exchanged between the fluid at temperature *T_k_* and the sensor at temperature *T* by Newton’s law [13,14]:(1)$$dQ=\alpha F\left(T_{k}-T\right)d$$

The amount of heat accumulated in the sensor [15]:(2)dQ=VρcdT=mcdT

Hence, the heat balance for a lumped parameter sensor is as given:(3)$$\alpha F\left(T_{k}-T\right)dt=mcdT$$
where:

*t*—time [s], 

*T*—sensor temperature [K], [°C], 

*m = V/ρ*—sensors mass [kg], 

*c*—specific heat of the sensor material [J/(kg·K)], 

*F*—heat transfer surface area [m^2^], 

*α*—heat transfer coefficient [W/(m^2^∙K)].

After substituting in Equation (3):(4)$$\tau =\frac{m\cdot c}{F\cdot \alpha }$$
a differential equation describing the first-order inertial term, can be obtained:(5)$$\frac{1}{\tau }\left(T_{k}-T\right)=\frac{dT}{dt}$$
or after separating the variables:(6)$$\frac{dT}{T_{k}-T}=-\frac{1}{\tau }dt$$

The solution to the differential Equation (6) with separated variables is found using the general integral method:(7)$$\int \frac{dT}{T_{k}-T}=\int -\frac{1}{\tau }dt$$

Solving the integral:(8)$$ln\left(T-T_{k}\right)=-\frac{1}{\tau }t+c$$

That is:(9)$$T-T_{k}=exp\left(-\frac{t}{\tau }+c\right)$$
(10)$$T-T_{k}=c_{1}exp\left(-\frac{t}{\tau }\right),\;where\;c_{1}=exp\left(c\right)$$
(11)$$T=T_{k}+c_{1}exp\left(-\frac{t}{\tau }\right)$$

Constant *c_1_* taking into account the internal conditions in Equation (11): *T = T*_0_, for the moment *t*_0_ = 0 s is equal to:(12)$$T_{0}=T_{k}+c_{1}exp\left(0\right)$$
(13)$$c_{1}={T_{0}-T}_{k}$$

Substituting Equation (13) into (11) the formula for the temperature of the sensor *T* with initial temperature *T*_0_ after immersion in a fluid at temperature *T_k_* can be obtained:(14)$$T=T_{k}-exp\left(-\frac{t}{\tau }\right)\left(T_{k}-T_{0}\right)$$

After transformation, the formula for the temperature of an ideal sensor is as given:(15)$$T=T_{0}+\left(T_{k}-T_{0}\right)\left(1-exp\left(-\frac{t}{\tau }\right)\right)$$

Or in dimensionless from:(16)$$\theta =\frac{{T-T}_{0}}{T_{k}-T_{0}}=\left(1-exp\left(-\frac{t}{\tau }\right)\right)$$

The parameter *τ* (4) is measured in time and is known as the time constant. It signifies the time taken by the sensor temperature (or indicated temperature) to reach 0.632 of the medium temperature jump value (as shown in Figure 1). The time constant depends on various factors such as the mechanical structure and dimensions of the sensor (*m*, *F*), the thermal properties of the material used to build the sensor (*c*), and the thermal properties of the fluid (*α*).

The time constant of a sensor is very different than its response time. In fact, the response time is exactly five times the time constant. Response time is the time for the sensor reading to reach 99.3% of the total step change in measurements, or in this case, the new temperature.

### 2.2. Determining the Time Constant for Real Sensors

Dynamic properties of real temperature sensors differ in many respects from those of an ideal sensor. The following reasons can be mentioned: the finite value of the thermal conductivity of the sensor material *λ*, changes in thermal transmittance *α* between the sensor and the tested medium occurring with changes in the temperature or flow speed of the medium, heat exchange between the temperature sensor and a medium other than the tested medium, the possibility of the sensor influencing the tested medium. (e.g., when the heat capacity of the sensor is much larger than the heat capacity of the medium), deformation of the temperature field of the tested medium. In real thermometers, it is possible to define the part that determines the indications of the device, e.g., in liquid thermometers, when the thickness of the glass layer is negligibly small, it will be the mass of liquid enclosed in the bulb. In uncovered resistance sensors with windings wound on an insulating body, this will be the sensor surface; in sheathed thermometers—the center (inside) of the sheath.

Depending on the ratio of times t_0.9_/t_0.5_ (Figure 2), real sensors are divided into the following types:

I. volumetric sensors: t_0.9_/t_0.5_ ≈ 3.32,

II. center-acting sensors: t_0.9_/t_0.5_ < 3.32,

III. surface acting sensors: t_0.9_/t_0.5_ > 3.32.

**Figure 2 sensors-24-00487-f002:**
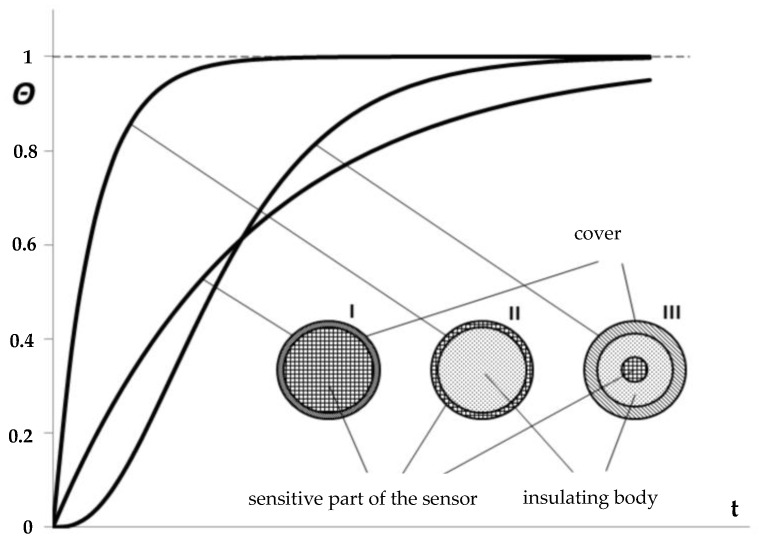
Relative temperature changes θ (16), depending on the type of sensor.

The time constants *t*_0.5_ and *t*_0.9_ determine the time after which the thermometer will indicate 50% and 90% of the temperature increase. 

The time constant of a temperature sensor is determined by the sensor’s response. It can be calculated analytically or determined empirically. Analytical methods involve identifying the dynamic parameters of the sensor based on equations describing the response to a unit step jump. These equations depend on the type of sensor.

The volumetric sensor (t_0.9_/t_0.5_ ≈ 3.32) is described assuming they constitute first-order inertial elements, and their properties are described by giving the value of the time constant *τ*. The transmittance of the sensor mapped in this way has the form of the transmittance of an ideal sensor:(17)$$G(s)=\frac{K_{T}}{1+s\cdot \tau }$$

The equation describing the sensor’s response to a unit step is:(18)$$T=T_{0}+\left(T_{k}-T_{0}\right)\left(1-e^{-\frac{t}{\tau }}\right)$$

The dynamic properties center-acting sensors (t_0.9_/t_0.5_ < 3.32) are described by:(a)specifying one-time constant τ determined in the same way as for volumetric sensors. This method is often used but not very accurate.(b)specifying delay time *τ*_0_ and time constant *τ*. The equivalent transmittance of the first-order inertial element, including delay, has the form:
(19)$$G(s)=K_{T}\frac{e^{-s\cdot \tau_{0}}}{\left(1+s\cdot \tau \right)}$$
and the response to a unit step is expressed with:(20)$$\left\{\begin{array}{c} T=T_{0}\;\;\;\;\;\;\;\;\;\;\;\;\;\;\;\;\;\;\;\;\;\;\;\;\;\;\;\;\;\;\;\;\;\;\;\;\;for\;\;\;\;\;\;\;\;t<\tau \\ T=T_{K}\left(1-e^{-\frac{t-t_{0}}{\tau }}\right)\;\;\;\;\;\;\;\;\;\;for\;\;\;\;\;\;\;\;t\geq \tau \end{array}\right.$$

(c)specifying the values of two time constants, *τ*_1_ and *τ*_2_, that correspond to the assumption, that the sensor consists of two first-order inertial elements connected in series, not interacting with each other. The equivalent transmittance of the sensor has the form:(21)$$G(s)=K_{T}\frac{1}{\left(1+s\cdot \tau_{1}\right)\cdot \left(1+s\cdot \tau_{2}\right)}$$
and the response to the step excitation is described by the relationship:(22)$$T=T_{0}+\left(T_{k}-T_{0}\right)\left[1-\frac{\tau_{1}}{\tau_{1}-\tau_{2}}e^{-\frac{t}{\tau_{1}}}+\frac{\tau_{2}}{\tau_{1}-\tau_{2}}e^{-\frac{t}{\tau_{2}}}\right]$$

Surface acting sensors (t_0.9_/t_0.5_ > 3.32) are described in two ways:(a)by specifying one time constant, treating the sensor as a first-order inertial element.(b)by specifying three time constants τ_1_, τ_2_ and τ_3_, using the sensor’s operator transfer function:
(23)$$G(s)=K_{T}\frac{1+s\tau_{3}}{\left(1+s\tau_{1}\right)\left(1+s\tau_{2}\right)}$$
the response to a unit jump is expressed as:(24)$$T=T_{k}\left[1-\frac{\tau_{1}-\tau_{3}}{\tau_{1}-\tau_{2}}e^{-\frac{t}{\tau_{1}}}+\frac{\tau_{2}-\tau_{3}}{\tau_{1}-\tau_{2}}e^{-\frac{t}{\tau_{2}}}\right]$$

Table 1 shows the values of time constants for sensors of various designs and manufacturers. It is clearly visible that the value of the time constant is influenced by heat transfer *α* (compare water and air). Additionally, protective covers impair the measurement dynamics.

## 3. Results

In order to analyze the time constants of contaminated temperature sensors and calibrate them, a test stand based on the Additel ADT875PC-155 dry well calibrator was used.

The tested sensors were PT500, Pt100, Ni100, and Cu50 in a 1H18N9T acid-resistant steel housing with a diameter of 6 mm. The article presents test results achieved for Pt500 sensors. The sensors were used in a clean version and with typical marine operational dirt. The temperature sensors were contaminated with paint, insulating tape, and epoxy resin. The sensors used to perform calibration are shown in Figure 3.

### 3.1. Typical Calibration

Thermometers and temperature sensors undergo calibration by the PN-EN IEC 60751 standard guidelines. Metrology centers adhere to standard procedures, which entail a minimum of three points being verified, with a waiting period of 3 min between temperature setting and reading. This time is fixed, and the time constant of the thermometer or sensor is not taken into account. As shown in Figure 1, the time constant is crucial when it comes to the accuracy of temperature measurement over a specific time period. For sensors with a short response time, 3 min is sufficient and often unnecessarily extended time to properly perform calibration. However, what will happen if a sensor with a longer time constant is taken into account?

The experimental test consisted of calibration in accordance with the principles adopted in routine tests carried out in metrology centers, taking into account the waiting time between the temperature setting in the furnace and the reading of 3 min. First, we place all the tested sensors in the dry well calibrator. The first measurement point is 0 °C. After the calibrator reaches this temperature, wait 3 min and then the measurement results of individual sensors are read. Then subsequent temperatures are set (40 °C, 60 °C, 120 °C). After reaching each of them, also wait 3 min before taking the measurement. The procedure is similar in the case of negative temperatures. The result of this procedure is shown in Figure 4.

The next stage was to perform calibration with the determination of the time constant to determine the waiting time and then to perform calibration. 

A study was carried out on the time constant of the sensors to a unit step. The jump size was 130 °C from room temperature (20 °C) to 150 °C. To determine the time constant, room temperature sensors were placed in an oven heated to 150 °C, and the results of the temperatures they measured were recorded with a sampling time of 1 s. The test results are presented in a chart (Figure 5).

As can be seen, operational contamination of the sensors significantly affects their time constants. It should be remembered that temperature sensors are used for various purposes. If the sensor is used as a part of regulation or automation systems, its response time is a key parameter and should be checked first. In case the permissible value resulting from the manufacturer’s specifications is exceeded, further calibration is pointless.

In the case of sensors used to control slowly changing parameters, where the response time is not as important as the accuracy of the measurement itself, too short a waiting time for the temperature to stabilize may affect the calibration result.

Figure 4 shows the calibration results considering four cases: clean PT500 sensors and those with typical operational contamination with 3 min waiting time.

As can be seen, not only does the operational contamination affect the value of the sensor’s time constant, but also the calibration result. Calibration performed with a 3-min waiting time confirmed the failure of the process in the case of 3 contaminated sensors. This conclusion could have been reached at the stage of checking the time constant, which would have significantly saved work and shortened both the time and costs of the entire calibration procedure.

If the time constant is not a key parameter, the waiting time should be adjusted after the temperature is set for a specific sensor. This time can be made dependent on the time constant measured during the first stage of calibration. Taking into account that the time signal allows the determination of when the sensor will reach a specific percentage of the set temperature. Also, the relative error of the measurement depending on the waiting time for the reading and the value of the unit jump can be determined.

Taking into account that each subsequent time constant causes a change in the sensor temperature by 63.2% of the difference between the current and final temperatures, this relationship can be described using the general formula for a geometric sequence [15]:(25)$$a_{n}=a_{k}q^{n-k}$$

The next words in the sequence can be described as:(26)$$a_{1}=0.632(T_{k}-T_{0})$$
(27)$$a_{2}=0.632(T_{k}-0.632T_{k})$$
where:

*T_k_*—final temperature, set and stabilized in the calibration oven [K], [°C].

*T*_0_—initial temperature [K], [°C].

Hence:(28)$$q=\frac{a_{2}}{a_{1}}=\frac{0.632(T_{k}-0.632T_{k})}{0.632T_{k}}=0.368$$

Taking Formulas (25) and (28) into account, the temperature indicated by the sensor after a specified time, which is a multiple of the time constant, the sensor can be described witch:(29)$$T_{n\tau }=T_{0}+\sum_{i=1}^{n}0.632\left(T_{k}-T_{0}\right)0.368^{\left(i-1\right)}$$
where:

*T_nτ_*—temperature indicated by the sensor after *n* time constants [K], [°C],

*n*—number of time constants.

The relative measurement error resulting from the time constant and the value of the unit stroke in relation to the permissible error of the sensor at a given temperature can be written as:(30)$$\delta =\frac{T_{k}-T_{n\tau }}{{∆}_{s}}\cdot 100$$
where:

Δ_s_—permissible error of the sensor (in the case of Pt500 it is 0.15 °C + 0.002 T_k_).

The relative error resulting from the time constant for the Pt500 sensor, after taking into Account (29), will be:(31)$$\delta_{Pt500}=\frac{T_{k}-T_{0}+\sum_{i=1}^{n}0.632\left(T_{k}-T_{0}\right)0.368^{\left(i-1\right)}}{0.15+0.002T_{k}}\cdot 100$$

Assuming the unit stroke variable of individual calibration steps as:(32)$${∆}_{T}=T_{k}-T_{0}$$
the final formula will take the form as given:(33)$$\delta_{Pt500}=\frac{{∆}_{T}+\sum_{i=1}^{n}0.632\left({∆}_{T}\right)0.368^{\left(i-1\right)}}{0.15+0.002T_{k}}\cdot 100$$

The relationship can also be presented graphically to facilitate the selection of the calibration time (Figure 6). Knowing the relationship between the sensor’s time constant, the permissible error, and the calibration time, the value of the relative error depending on the calibration time can be determined.

To facilitate the use of the chart in everyday practice, it can also be presented in 2D form.

### 3.2. Calibration of Real Objects Using the Proposed Method

Using the results of the time constant measurement (Figure 5) and Formula (33), it was assumed that, for the calibrated sensors, the relative measurement error resulting from the waiting time for the temperature to stabilize should not exceed 5%.

In order to check the assumptions presented above, a full calibration of the sensors was carried out for Pt500 sensors in 1H18N9T acid-resistant steel housings with a diameter of 6 mm, clean and operationally contaminated with time constants: 26, 40, 48, and 57 [s].

Using the graph in Figure 7, it was determined that with a temperature jump of 40 °C and assuming that the error resulting from temperature stabilization does not exceed 5%, the temperature stabilization time before obtaining the calibration result should be 7.8*τ*.

For such assumptions, full calibration of the sensors was performed in accordance with the currently applicable procedure, i.e., with a stabilization time of 3 min. Next, a recommended temperature stabilization time of 7.8*τ*. The results are presented in the form of graphs below. The uncertainty estimation was carried out in accordance with the provisions of the EA-4/02 M: 2022 “Evaluation of the Uncertainty of Measurement in calibration” standard [17] (Figure 8).

## 4. Discussion

As a result of the mathematical analyses, the relationship (33) was obtained, which allows for determining the waiting time for the temperature stabilization of the sensor depending on its time constant. Using the measurement of the sensor’s time constant as the first calibrated parameter allows the verification of the need to conduct a full test. In the case of sensors measuring operational parameters for which it is required to maintain the dynamic parameters of the sensor, the detection of an extended time constant eliminates it from further operation, and carrying out a full metrological verification procedure is pointless.

In the case of sensors used to measure quantities for which the dynamic parameters of the sensor are not of decisive importance, determining the time constant should also be the first step in the calibration process. Based on the measured time constant of the sensor and the Relationship (33), the preferred waiting time for the measurements from the tested sensor to stabilize can be determined. It turns out that, despite contamination and a significant increase in the time constant, the sensors remain metrologically efficient, provided that the appropriate time is used for temperature stabilization of the sensor.

The proposed solution allows for the optimization of the calibration process of resistance temperature sensors, which certainly affects both its timespan and costs.

## Figures and Tables

**Figure 1 sensors-24-00487-f001:**
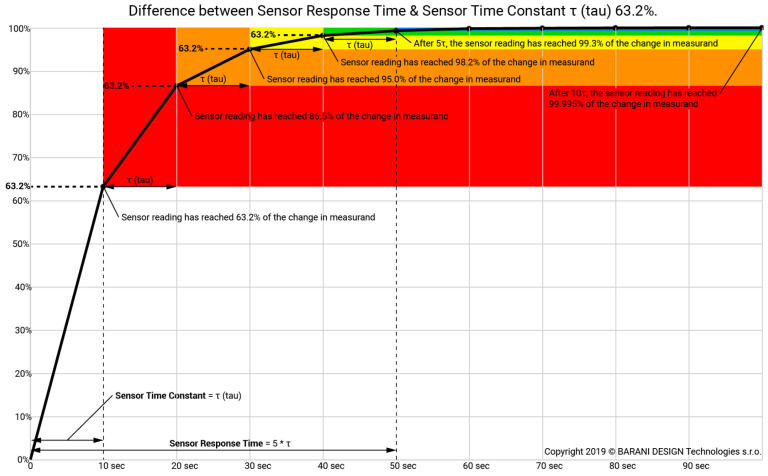
How sensor response time relates to the sensor time constant [16].

**Figure 3 sensors-24-00487-f003:**
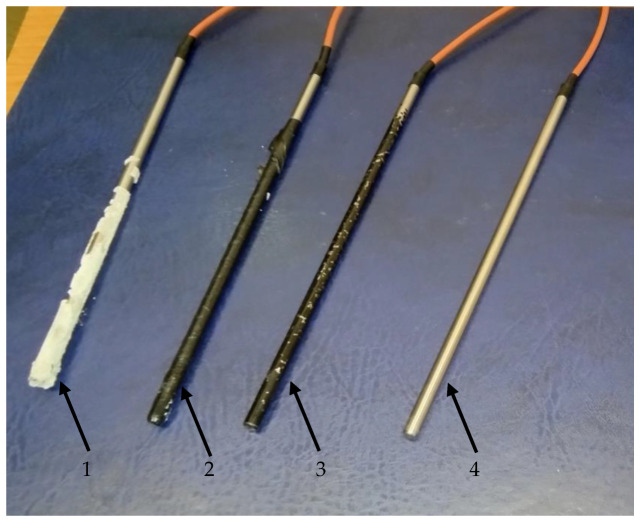
Contaminated sensors subject to calibration 1—epoxy resin, 2—insulating tape, 3—paint, 4—clean sensor.

**Figure 4 sensors-24-00487-f004:**
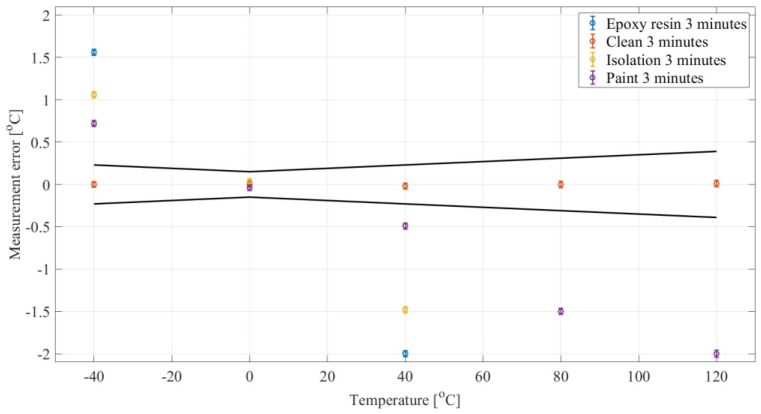
Typical calibration results.

**Figure 5 sensors-24-00487-f005:**
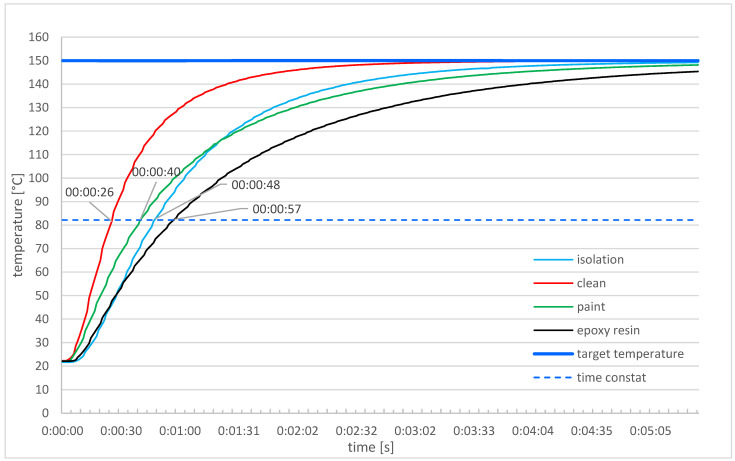
Time constants for different sensors.

**Figure 6 sensors-24-00487-f006:**
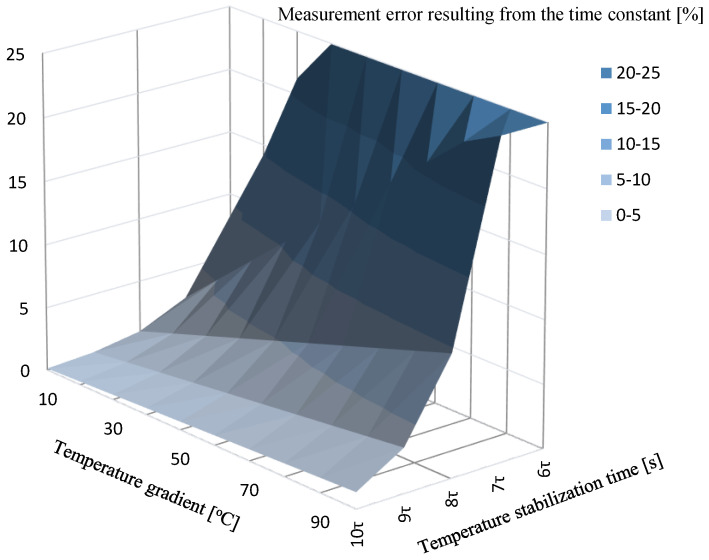
Measurement error resulting from the time constant and temperature gradient.

**Figure 7 sensors-24-00487-f007:**
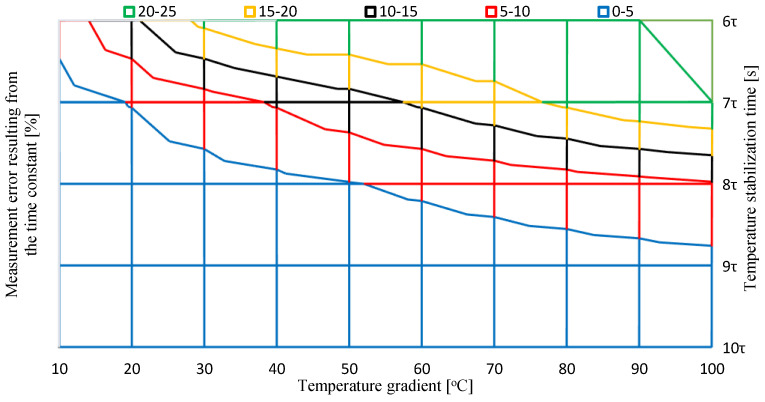
Measurement error resulting from the time constant and temperature gradient 2D version.

**Figure 8 sensors-24-00487-f008:**
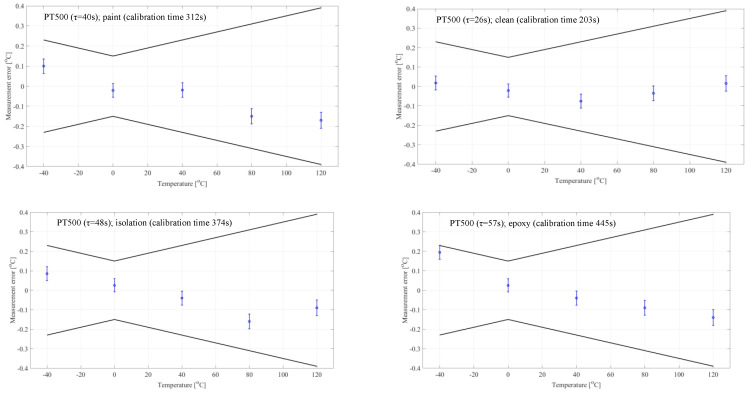
Calibration results for different sensors with different stabilization times.

**Table 1 sensors-24-00487-t001:** Time constants for different types of sensors.

Sensor with Cover Dimensions [mm] (Pipe Diameter × Wall Thickness)	Time Constants Water t_0.5_	Time Constants Water t_0.9_	Time Constants Air t_0.5_	Time Constants Air t_0.9_
3 × 0.25	7	19	35	95
4 × 0.35	12	32	45	125
5 × 0.35	17	45	70	190
6 × 0.35	22	61	80	235
6 × 0.50	12	55	90	260
8 × 0.60	20	85	125	360
10 × 1.5	27	72	105	310
10 × 1.6	35	100	150	400
12 × 1.5	45	155	180	450
12 × 2	51	138	165	420
15 × 1.5	57	170	190	490
22 × 2	130	480	480	1200
	ceramic cover			
6 × 1	20	55	75	180
10 × 2	30	92	100	270
15 × 2	42	125	220	580

## Data Availability

Data are contained within the article.

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
