# Peer review of "Time Constant as the Main Component to Optimize the Resistance Temperature Sensors’ Calibration Process"

_sensors, 2024, doi:10.3390/s24020487_

Round 1

Reviewer 1 Report

Comments and Suggestions for Authors

The calibration of temperature sensors, especially on-site recalibration during service, is a very important issue. This study provides an interesting concise solution for those sensors whose covers are partially damaged, dirty, or covered with something.

Through theoretical analysis and deduction, this study proposed to use the on-site measurement value of the time constant of the measured sensor as the first calibration parameter. This method is helpful to conduct a reasonable full calibration test according to actual needs during the dynamic calibration procedure.

In addition, please explain or provide the reasons and basis for selecting the cover dimensions in Table 1.

Comments on the Quality of English Language

The English of the manuscript is good except for minor errors. Such as “Cover” in Figure 2 didn’t show fully.

Author Response

Thank you very much for your comments. All of them were taken into account in the updated version of the article.

The dimensions of the sensors shown in Table 1 are typical dimensions of sensors widely used in the industry. This demonstration was intended to show the influence of the housing on the sensor's time constant.

The word "cover" is fully visible in Figure 2.

Reviewer 2 Report

Comments and Suggestions for Authors

The article "Time constant as the main element of the resistance temperature sensors' calibration process' optimisation" is an original and interesting paper. It is suitable for publication in the journal Sensors, but needs a number of improvements.

My comments on the article:

1. In the References section, the formatting style of literature items should be standardized in accordance with the journal's guidelines. The titles of articles in items 5 and 7 of the References chapter are missing.

2. The literature review in the Introduction was done very vaguely. It is necessary to present in a few sentences one by one all the studies cited (items [1-7]). Studies on the dynamic properties of thermoelectric thermometers and the influence of the heat transfer coefficient on the value of the time constant are also presented in works carried out at the Cracow University of Technology, e.g.: 

Marek MAJDAK, Magdalena JAREMKIEWICZ: "Analysis of thermocouples time constants as a function of fluid velocity", Measurement Automation Monitoring, Jan. 2015, vol. 61, no.01.

3. The text of the paper needs minor corrections, i.e.:

- if a colon is already inserted before the equations, it should be after the cited literature, and not before as, for example, in lines 86, 87,

- the multiplication sign "×" in equations is unnecessary,

- units in the description of symbols need editing (lines 92, 95, 96),

- line 255 is missing a unit in the symbol description. 

The work should be carefully checked for such shortcomings.

4. In line 96, the symbol α should be described as "heat transfer coefficient" (this is a term commonly used in English-language literature).

5. In line 106, it should be clearly shown that t0 = 0s.

6. In lines 113-114 it should be: "...to reach 0.632 of the medium temperature jump value...".

7. The sentences in lines 112-114 are the same as the sentences in lines 120-122.

8. The paper lacks an explanation of the symbols t0.5 and t0.9.

9. Why is the time constant τ3 missing in equation (24)? Is equation (24) complete?

10. The Authors of the paper should describe the method of the tests conducted, the results of which are presented in Section 3.1.

11. Figure 3 is insufficiently described. The description in the form of mental abbreviations is unacceptable, it should be unambiguous. What seems obvious to the Author of the paper, to other readers is not necessarily so. What do the descriptions mean: "paint", "epoxy resin", "isolation"? What does the temperature mean by the description and graph "temp"? It also appears from the drawing as presented in the article that the time constant is represented by the blue dashed line and is approximately 95ËšC, which is clearly not the truth or the Authors' intention. In contrast, the specific values of the time constants of the individual thermometers are not described in any way. The description is similarly unclear in Figure 4.

12. In line 263, there should be a reference to equation (29).

13. In Figure 5, the description of the variables "Measurement error resulting from the time constant" should be at the top of the graph where the error ranges are indicated by colors.

14. In my opinion, the format of the time constant values in lines 280 and 281 should be changed as in the following examples: is 0:31 s, should be 31 s, is 1:14 s, should be 74 s.

Author Response

Thank you very much for your comments. All of them were taken into account in the updated version of the article.

The references section has been improved.

The literature was revised

Lines 86, 87, 92, 95, 96, 106, 255, 263, 280, 281 have been corrected as suggested.

The multiplication sign has been removed from the equations.

Symbols t0.5, t0.9 are now described on lines 145,146.

The description of Figure 3 has been corrected (now Figure 4). Indeed, the "time constant" line at 950C was an editorial mistake - corrections have been made. A more detailed way of testing the test has been added. A description and drawing of individual pollutants have been added.

Equation (24) has been corrected.

Figure 3 (now Figure 5) have been corrected as suggested.

Reviewer 3 Report

Comments and Suggestions for Authors

Dear Authors,

Your article is devoted to an important branch of metrology - the calibration of temperature sensors.

However, please consider the following comments.

1. In the title (line 2) there is a wording: "Time constant as the main element...". I am confused that the authors refer to the time constant as an element.

2. In the introduction (lines 30 to 33) there is a mix of terms: "...physical quantities such as pressure, flows,...". It is not clear to me how the authors classify flow as a physical quantity.

3. In the same sentence (lines 30 to 33) it is stated that when thermal processes dominate in power equipment, pressure (for example) changes slowly and therefore measurement accuracy can be neglected. I cannot agree with this statement. The authors need to carefully reword the sentence.

4. I have the impression that all literature references are only in the introduction. The reference list itself of 12 sources looks unconvincing.

5. In line 47 the authors mentioned "response time". But if, as the authors themselves say below, this value is simply 5 times greater than the time constant, why check it?

6. The sentence (lines 51 and 52) suggests that the authors will try to verify the effect of pollutants. How does this statement fit with the title and abstract?

7. In lines 68 and 69, the authors say that the Dirac pulse is difficult to do in practice. It is hard to agree because there are fast pulsed lasers. They are just right for non-stationary calibration of temperature sensors.

8. It is strange that the authors took Newton's cooling law from paper 10 and not from another respected classical textbook (lines 85 and 86).

9. In line 85 the authors mentioned only convection. Aren't temperature sensors also subject to radiation? It is radiation that can give a much larger error in measurement.

10. Figure 1 is taken from source 12. Source 12 is the website of a company selling weather stations. I think a more authoritative source is needed for a scientific paper.

11. Line 142 refers to Figure 2 "(Fig.2)". All other figures are referred to by the authors with the full word Figure. I think it is advisable to follow uniformity.

12. In lines 200 to 205, the authors describe the equipment: calibrator and sensors. However, they do not say anything about the procedure of the experiments. Were the sensors simply dropped into the calibrator? What are the losses when putting a cold sensor into the calibrator? Nothing is clear here.

13. In line 212, there is a dot sign between the words "temperature" and "measurement".

14. In line 216, the authors refer to Figure 3 and write "response time". However, the title of the figure states time constant.

15. The time scale in Figure 3 is signed in seconds, but the axis has numbers as 00:02:00 for example. Is it 2 minutes? There is a discrepancy between the numbers and the caption.

16. The authors write the term "error" everywhere in the paper. Today it is recommended to evaluate experimental results for uncertainty of measurement (Type A and Type B), e.g. according to the "Guide to the Expression of Uncertainty of Measurement" (GUM). ISO/IEC Guide 98-1: 2009, Uncertainty of measurement - Part 1: Introduction to the expression of uncertainty of measurement, IDT. More detailed information can be found at https://www.bipm.org/en/publications/guides/.

However, there is no section on uncertainty in the manuscript.

17. In lines 282-284, the authors refer to and discuss Figure 8. But there are only 6 figures in the article. The Figure 6 itself has very a small font.

18. In line 299, the authors introduce a new term, "waiting time". What are the authors studying anyway?

19. The conclusion (lines 313 and 314) looks unproven.

Summarising my review, I make the following conclusion: the manuscript requires substantial revision.

Best regards, reviewer.

Author Response

Thank you very much for your comments. All of them were taken into account in the updated version of the article.

  1. The title of the article has been changed to “Time constant as the main component of the resistance temperature sensors’ calibration process’ optimization”
  2. In line 30 (now 31), the physical quantity has been replaced by a physical parameter
  3. The literature was revised
  4. Mentioned sentence was reworded according to given suggestions
  5. In line 47 (now 53), the response time has been replaced by a time constant. In fact, the calibration process measures the time constant as described later in the article.
  6. Testing the impact of contamination using a time constant allows determining the appropriateness of calibration. In the case of an increased time constant, it allows for the appropriate selection of the temperature stabilization time during calibration. Both of these aspects allow for optimization of the entire calibration process.
  7. The part about the difficulties in practically producing a Dirack impulse has been removed
  8. The literature reference in lines 85, 86 (now 88) has been supplemented.
  9. Line 85 (now 88) presents the basic theoretical assumptions. Indeed, temperature sensors read temperature through both convection, conduction and radiation. The article only presents the impact of pollutants on the change in the sensor time constant, regardless of the type of heat transfer. The phrase "by convection" has been removed.
  10. Information presented in the Figure 1 are widely and commonly known by scientific community. It was taken form manufacturer’s website, what means that the device must have been tested and given information are verified.
  11. Line 142 has been corrected as suggested.
  12. A detailed description of the measurement procedure has been added in section 3.1.
  13. Line 212 has been corrected as suggested.
  14. Line 216 has been corrected as suggested.
  15. The timeline in Figure 3 (now Figure 5) has been scaled for clarity
  16. In lines 284-286 it is noted that the measurement uncertainty estimation was performed in accordance with the recommendations of document EA-4/02 M: 2022 "Evaluation of the Uncertainty of Measurement in calibration". There is no separate chapter on uncertainty estimation in the article because it is a standard activity performed in the calibration process and regardless of the degree of contamination of the sensors, it is performed in the same way.
  17. On line 278, "Figure 8" has been replaced by "Figure 5"
  18. The term "waiting time" is explained in lines 309-311. This is not a new quantity being tested. "Waiting time" is the time between the stabilization of the furnace temperature and the measurement of the temperature.

Round 2

Reviewer 3 Report

Comments and Suggestions for Authors

Dear authors,

Thank you very much for your comments on my questions! 

I found only one point in the new version of your manuscript that I think can be improved:

The font size of some figures (e.g. Figure 8, lines 325-329) seems to me to be too small. 

Kind regards and good luck with your future research,

Reviewer.